# What Are People Asking About COVID-19?
# A Question Classification Dataset

**Jerry Wei**♠ **Chengyu Huang**♦ **Soroush Vosoughi**♥ **Jason Wei**♥

♠Protago Labs ♦International Monetary Fund ♥Dartmouth College
`jerry.weng.wei@protagolabs.com`
`huangchengyu24@gmail.com`
`{soroush,jason.20}@dartmouth.edu`

## Abstract

We present COVID-Q, a set of 1,690 questions about COVID-19 from 13 sources, which we annotate into 15 question categories and 207 question clusters. The most common questions in our dataset asked about transmission, prevention, and societal effects of COVID, and we found that many questions that appeared in multiple sources were not answered by any FAQ websites of reputable organizations such as the CDC and FDA. We post our dataset publicly at `https://github.com/JerryWei03/COVID-Q`.

For classifying questions into 15 categories, a BERT baseline scored 58.1% accuracy when trained on 20 examples per category, and for a question clustering task, a BERT + triplet loss baseline achieved 49.5% accuracy. We hope COVID-Q can help either for direct use in developing applied systems or as a domain-specific resource for model evaluation.

## 1 Introduction

A major challenge during fast-developing pandemics such as COVID-19 is keeping people updated with the latest and most relevant information. Since the beginning of COVID, several websites have created frequently asked questions (FAQ) pages that they regularly update. But even so, users might struggle to find their questions on FAQ pages, and many questions remain unanswered. In this paper, we ask—what are people really asking about COVID, and how can we use NLP to better understand questions and retrieve relevant content?

We present COVID-Q, a dataset of 1,690 questions about COVID from 13 online sources. We annotate COVID-Q by classifying questions into 15 general *question categories*[1] (see Figure 1) and by grouping questions into *question clusters*, for which all questions in a cluster ask the same thing

---

[1]We do not count the "other" category.

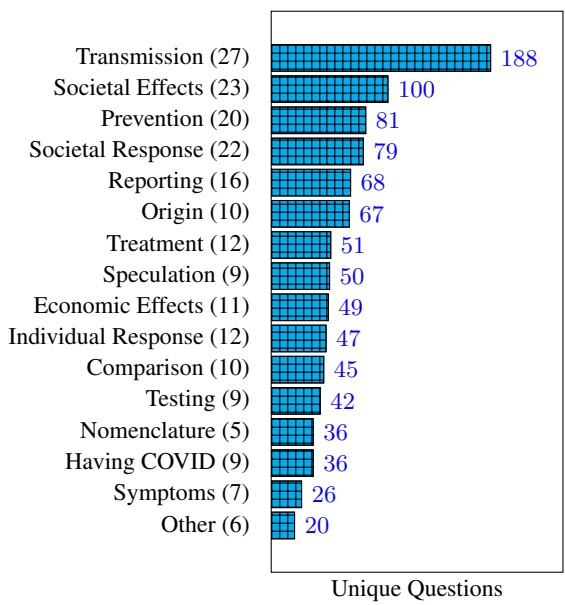

Figure 1: Question categories in COVID-Q, with number of question clusters per category in parentheses.

and can be answered by the same answer, for a total of 207 clusters. Throughout §2, we analyze the distribution of COVID-Q in terms of question category, cluster, and source.

COVID-Q facilitates several question understanding tasks. First, the question categories can be used for a vanilla text classification task to determine the general category of information a question is asking about. Second, the question clusters can be used for retrieval question answering (since the cluster annotations indicate questions of same intent), where given a new question, a system aims to find a question in an existing database that asks the same thing and returns the corresponding answer (Romeo et al., 2016; Sakata et al., 2019). We provide baselines for these two tasks in §3.1 and §3.2. In addition to directly aiding the development of potential applied systems, COVID-Q could also serve as a domain-specific resource for evaluating NLP models trained on COVID data.

| Source | Total | Questions Multi-q-cluster | Questions Single-q-cluster | Answers | Questions Removed |
|---|---|---|---|---|---|
| Quora | 675 | 501 (74.2%) | 174 (25.8%) | 0 | 374 |
| Google Search | 173 | 161 (93.1%) | 12 (6.9%) | 0 | 174 |
| github.com/deepset-ai/COVID-QA | 124 | 55 (44.4%) | 69 (55.6%) | 124 | 71 |
| Yahoo Search | 94 | 87 (92.6%) | 7 (7.4%) | 0 | 34 |
| *Center for Disease Control | 92 | 51 (55.4%) | 41 (44.6%) | 92 | 1 |
| Bing Search | 68 | 65 (95.6%) | 3 (4.4%) | 0 | 29 |
| *Cable News Network | 64 | 48 (75.0%) | 16 (25.0%) | 64 | 1 |
| *Food and Drug Administration | 57 | 33 (57.9%) | 24 (42.1%) | 57 | 3 |
| Yahoo Answers | 28 | 13 (46.4%) | 15 (53.6%) | 0 | 23 |
| *Illinois Department of Public Health | 20 | 18 (90.0%) | 2 (10.0%) | 20 | 0 |
| *United Nations | 19 | 18 (94.7%) | 1 (5.3%) | 19 | 6 |
| *Washington DC Area Television Station | 16 | 15 (93.8%) | 1 (6.2%) | 16 | 0 |
| *Johns Hopkins University | 11 | 10 (90.9%) | 1 (9.1%) | 11 | 1 |
| Author Generated | 249 | 249 (100.0%) | 0 (0.0%) | 0 | 0 |
| Total | 1,690 | 1,324 (78.3%) | 366 (21.7%) | 403 | 717 |

Table 1: Distribution of questions in COVID-Q by source. The reported number of questions excludes vague and nonsensical questions that were removed. Multi-q-cluster: number of questions that belonged to a question cluster with at least two questions; Single-q-cluster: number of questions that belonged to a question cluster with only a single question (no other question in the dataset asked the same thing). * denotes FAQ page sources.

## 2 Dataset Collection and Annotation

**Data collection.** In May 2020, we scraped questions about COVID from thirteen sources: seven official FAQ websites from recognized organizations such as the Center for Disease Control (CDC) and the Food and Drug Administration (FDA), and six crowd-based sources such as Quora and Yahoo Answers. Table 1 shows the distribution of collected questions from each source. We also post the original scraped websites for each source.

**Data cleaning.** We performed several preprocessing steps to remove unrelated, low-quality, and nonsensical questions. First, we deleted questions unrelated to COVID and vague questions with too many interpretations (e.g., "Why COVID?"). Second, we removed location-specific and time-specific versions of questions (e.g., "COVID deaths in New York"), since these questions do not contribute linguistic novelty (you could replace "New York" with any state, for example). Questions that only targeted one location or time, however, were not removed—for instance, "Was China responsible for COVID?" was not removed because no questions asked about any other country being responsible for the pandemic. Finally, to minimize occurrences of questions that trivially differ, we removed all punctuation and replaced synonymous ways of saying COVID, such as "coronavirus," and "COVID-19" with "covid." Table 1 also shows the number of removed questions for each source.

| Question Cluster [#Questions] (Category) | Example Questions |
|---|---|
| Pandemic Duration [28] (Speculation) | "Will COVID ever go away?" "Will COVID end soon?" "When COVID will end?" |
| Demographics: General [26] (Transmission) | "Who is at higher risk?" "Are kids more at risk?" "Who is COVID killing?" |
| Survivability: Surfaces [24] (Transmission) | "Does COVID live on surfaces?" "Can COVID live on paper?" "Can COVID live on objects?" |

Table 2: Most common question clusters in COVID-Q.

**Data annotation.** We first annotated our dataset by grouping questions that asked the same thing together into question clusters. The first author manually compared each question with existing clusters and questions, using the definition that two questions belong in the same cluster if they have the same answer. In other words, two questions matched to the same question cluster if and only if they could be answered with a common answer. As every new example in our dataset is checked against all existing question clusters, including clusters with only one question, the time complexity for annotating our dataset is $O(n^2)$, where $n$ is the number of questions.

After all questions were grouped into question clusters, the first author gave each question cluster with at least two questions a name summarizing the questions in that cluster, and each question cluster was assigned to one of 15 question categories (as

shown in Figure 1), which were conceived during a thorough discussion with the last author. In Table 2, we show the question clusters with the most questions, along with their assigned question categories and some example questions. Figure 2 shows the distribution of question clusters.

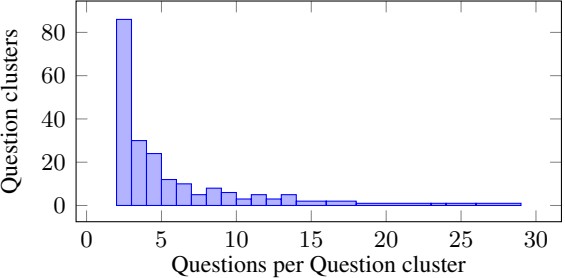

Figure 2: Number of questions per question cluster for clusters with at least two questions. All questions in a question cluster asked roughly the same thing. 120 question clusters had at least 3 questions per cluster, 66 clusters had at least 5 questions per cluster, and 22 clusters had at least 10 questions per cluster.

**Annotation quality.** We ran the dataset through multiple annotators to improve the quality of our annotations. First, the last author confirmed all clusters in the dataset, highlighting any questions that might need to be relabeled and discussing them with the first author. Of the 1,245 questions belonging to question clusters with at least two questions, 131 questions were highlighted and 67 labels were modified. For a second pass, an external annotator similarly read through the question cluster labels, for which 31 questions were highlighted and 15 labels were modified. Most modifications involved separating a single question cluster that was too broad into several more specific clusters.

For another round of validation, we showed three questions from each of the 89 question clusters with $N_{cluster} \geq 4$ to three Mechanical Turk workers, who were asked to select the correct question cluster from five choices. The majority vote from the three workers agreed with our ground-truth question-cluster labels 93.3% of the time. The three workers unanimously agreed on 58.1% of the questions, for which 99.4% of these unanimous labels agreed with our ground-truth label. Workers were paid $0.07 per question.

Finally, it is possible that some questions could fit in several categories—of 207 clusters, 40 arguably mapped to two or more categories, most frequently the transmission and prevention categories. As this annotation involves some degree of subjectivity, we post formal definitions of each

question category with our dataset to make these distinctions more transparent.

**Single-question clusters.** Interestingly, we observe that for the CDC and FDA frequently asked questions websites, a sizable fraction of questions (44.6% for CDC and 42.1% for FDA) did not ask the same thing as questions from any other source (and therefore formed *single-question clusters*), suggesting that these sources might want adjust the questions on their websites to question clusters that were seen frequently in search engines such as Google or Bing. Moreover, 54.2% of question clusters that had questions from at least two non-official sources went unanswered by an official source. In the Supplementary Materials, Table 7 shows examples of these questions, and conversely, Table 8 shows CDC and FDA questions that did not belong to the same cluster as any other question.

## 3 Question Understanding Tasks

We provide baselines for two tasks: *question-category classification*, where each question belongs to one of 15 categories, and *question clustering*, where questions asking the same thing belong to the same cluster.

As our dataset is small when split into training and test sets, we manually generate an additional *author-generated* evaluation set of 249 questions. For these questions, the first author wrote new questions for question clusters with 4 or 5 questions per cluster until those clusters had 6 questions. These questions were checked in the same fashion as the real questions. For clarity, we only refer to them in §3.1 unless explicitly stated.

### 3.1 Question-Category Classification

The *question-category classification* task assigns each question to one of 15 categories shown in Figure 1. For the train-test split, we randomly choose 20 questions per category for training (as the smallest category has 26 questions), with the remaining questions going into the test set (see Table 3).

| Question Categories | 15 |
|---|---|
| Training Questions per Category | 20 |
| Training Questions | 300 |
| Test Questions (Real) | 668 |
| Test Questions (Generated) | 238 |

Table 3: Data split for *question-category classification*.

We run simple BERT (Devlin et al., 2019) feature-extraction baselines with question representations obtained by average-pooling. For this

task, we use two models: (1) SVM and (2) cosine-similarity based $k$-nearest neighbor classification ($k$-NN) with $k = 1$. As shown in Table 4, the SVM marginally outperforms $k$-NN on both the real and generated evaluation sets. Since our dataset is small, we also include results from using data augmentation (Wei and Zou, 2019). Figure 4 (Supplementary Materials) shows the confusion matrix for BERT-feat: SVM + augmentation for this task.

| Model | Real Q | Generated Q |
|---|---|---|
| BERT-feat: $k$-NN | 47.8 | 52.1 |
| + augmentation | 47.3 | 52.5 |
| BERT-feat: SVM | 52.2 | 53.4 |
| + augmentation | 58.1 | 58.8 |

Table 4: Performance of BERT baselines (accuracy in %) on *question-category classification* with 15 categories and 20 training examples per category.

## 3.2 Question Clustering

Of a more granular nature, the *question clustering* task asks, given a database of known questions, whether a new question asks the same thing as an existing question in the database or whether it is a novel question. To simulate a potential applied setting as much as possible, we use all questions clusters in our dataset, including clusters containing only a single question. As shown in Table 5, we make a 70%–30% train–test split by class.[2]

| | |
|---|---|
| Training Questions | 920 |
| Training Clusters | 460 |
| Test Questions | 437 |
| Test Clusters | 320 |
| Test Questions from multi-q-clusters | 323 |
| Test Questions from single-q-clusters | 114 |

Table 5: Data split for *question clustering*.

In addition to the $k$-NN baseline from §3.1, we also evaluate a simple model that uses a triplet loss function to train a two-layer neural net on BERT features, a method introduced for facial recognition (Schroff et al., 2015) and now used in NLP for few-shot learning (Yu et al., 2018) and answer selection (Kumar et al., 2019). For evaluation, we compute a single accuracy metric that requires a question to be either correctly matched to a cluster in the database or to be correctly identified as a novel question. Our baseline models use thresholding to determine

---

[2]For clusters with two questions, one question went into the training set and one into the test set. 70% of single-question clusters went into the training set and 30% into the test set.

| Model | Accuracy (%) | |
|---|---|---|
| | Top-1 | Top-5 |
| BERT-feat: $k$-NN | 39.6 | 58.8 |
| + augmentation | 39.6 | 59.0 |
| BERT-feat: triplet loss | 47.7 | 66.9 |
| + augmentation | 49.5 | 69.4 |

Table 6: Performance of BERT baselines on *question clustering* involving 207 clusters.

whether questions were in the database or novel. Table 6 shows the accuracy from the best threshold for both these models, and Supplementary Figure 3 shows their accuracies for different thresholds.

## 4 Discussion

**Use cases.** We imagine several use cases for COVID-Q. Our question clusters could help train and evaluate retrieval-QA systems, such as `covid.deepset.ai` or `covid19.dialogue.co`, which, given a new question, aim to retrieve the corresponding QA pair in an existing database. Another relevant context is query understanding, as clusters identify queries of the same intent, and categories identify queries asking about the same topic. Finally, COVID-Q could be used broadly to evaluate COVID-specific models—our baseline (Hugging-face's `bert-base-uncased`) does not even have *COVID* in the vocabulary, and so we suspect that models pre-trained on scientific or COVID-specific data will outperform our baseline. More related areas include COVID-related query expansion, suggestion, and rewriting.

**Limitations.** Our dataset was collected in May 2020, and we see it as a snapshot in time of questions asked up until then. As the COVID situation further develops, a host of new questions will arise, and the content of these new questions will potentially not be covered by any existing clusters in our dataset. The question categories, on the other hand, are more likely to remain static (i.e., new questions would likely map to an existing category), but the current way that we came up with the categories might be considered subjective—we leave that determination to the reader (refer to Table 9 or the raw dataset on Github). Finally, although the distribution of questions per cluster is highly skewed (Figure 2), we still provide them at least as a reference for applied scenarios where it would be useful to know the number of queries asking the same thing (and perhaps how many answers are needed to answer the majority of questions asked).

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

# 5 Supplementary Materials

## 5.1 Question Clustering Thresholds

For the question clustering task, our models used simple thresholding to determine whether a question matched an existing cluster in the database or was novel. That is, if the similarity between a question and its most similar question in the database was lower than some threshold, then the model predicted that it was a novel question. Figure 3 shows the accuracy of the $k$-NN and triplet loss models at different thresholds.

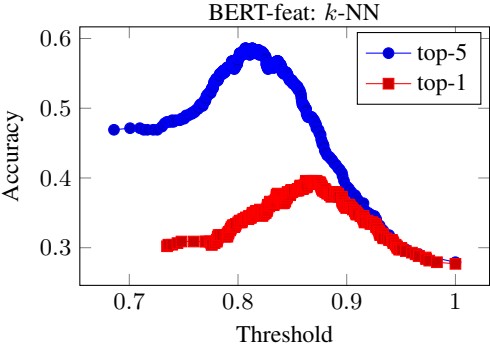

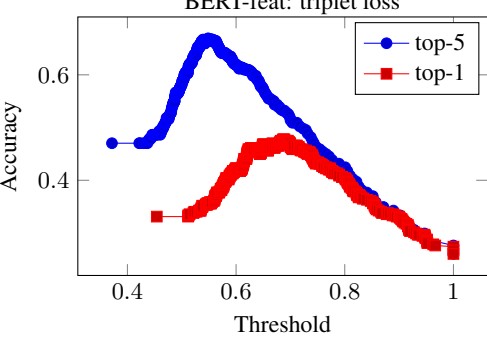

Figure 3: Question clustering accuracy for $k$-NN and triplet loss models at different thresholds. If a given test question had a similarity that was less than the threshold, then it was classified as a novel question (i.e., not in the database of known questions). When the threshold was too high, performance dropped because too many questions were classified as novel. When the threshold was too low, performance dropped because the model attempted to match too many test questions to existing clusters in the database.

## 5.2 Question-Category Classification Error Analysis

Figure 4 shows the confusion matrix for our SVM classifier on the question-category classification task on the test set of real questions. Categories that were challenging to distinguish were *Transmission* and *Having COVID* (34% error rate), and *Having COVID* and *Symptoms* (33% error rate).

## 5.3 Further Dataset Details

**Question mismatches.** Table 7 shows example questions from at least two non-official sources that went unanswered by an official source. Table 8 shows example questions from the FDA and CDC FAQ websites that did not ask the same thing as any other questions in our dataset.

| Question Cluster | $N_{cluster}$ | Example Questions |
|---|---|---|
| Number of Cases | 21 | "Are COVID cases dropping?" "Have COVID cases peaked?" "Are COVID cases decreasing?" |
| Mutation | 19 | "Has COVID mutated?" "Did COVID mutate?" "Will COVID mutate?" |
| Lab Theory | 18 | "Was COVID made in a lab?" "Was COVID manufactured?" "Did COVID start in a lab?" |

Table 7: Questions appearing in multiple sources that were unanswered by official FAQ websites.

**Example questions.** Table 9 shows example questions from each of the 15 question categories.

**Corresponding answers.** The FAQ websites from reputable sources (denoted with * in Table 1) provide answers to their questions, and so we also provide them as an auxiliary resource. Using these answers, 23.8% of question clusters have at least one corresponding answer. We caution against using these answers in applied settings, however, because information on COVID changes rapidly.

**Additional data collection details.** In terms of how questions about COVID were determined, for FAQ websites from official organizations, we considered all questions, and for Google, Bing, Yahoo, and Quora, we searched the keywords "COVID" and "coronavirus."

As for synonymous ways of saying COVID, we considered "SARS-COV-2," "coronavirus," "2019-nCOV," "COVID-19," and "COVID19."

**Other COVID-19 datasets.** We encourage researchers to also explore other COVID-19 datasets: tweets streamed since January 22 (Chen et al., 2020), location-tagged tweets in 65 languages (Abdul-Mageed et al., 2020), tweets of COVID symptoms (Sarker et al., 2020), a multi-lingual Twitter and Weibo dataset (Gao et al., 2020), an Instagram dataset (Zarei et al., 2020), emotional responses to COVID (Kleinberg et al., 2020), and annotated research abstracts (Huang et al., 2020).

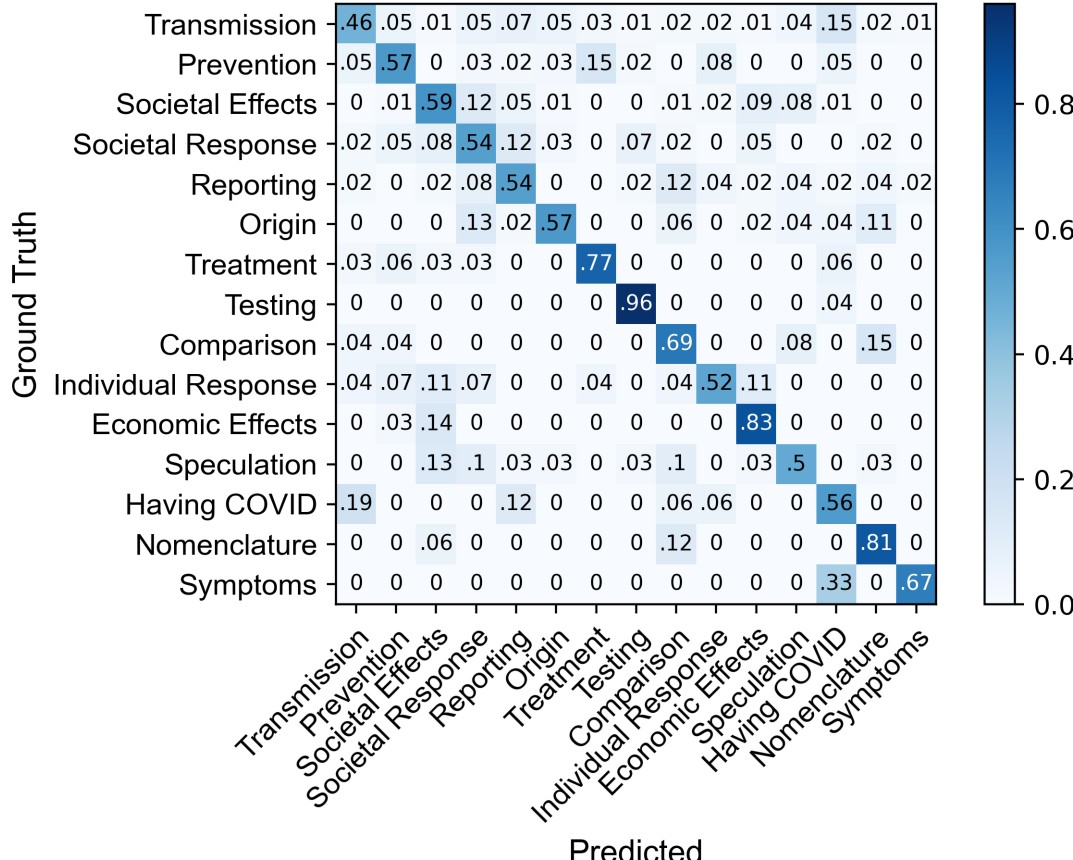

Figure 4: Confusion matrix for BERT-feat: SVM predictions on the question-category classification task.

| Question | Food and Drug Administration
Closest Matches from BERT |
|---|---|
| "Can I donate convalescent plasma?" | "Why is convalescent plasma being investigated to treat COVID?"
"Can I make my own hand sanitizer?"
"What are suggestions for things to do in the COVID quarantine?" |
| "Where can I report websites selling fraudulent medical products?" | "What kind of masks are recommended to protect healthcare workers from COVID exposure?"
"Where can I get tested for COVID?"
"How do testing kits for COVID detect the virus?" |
| Question | Center for Disease Control
Closest Matches from BERT |
| "What is the difference between cleaning and disinfecting?" | "How effective are alternative disinfection methods?"
"Why has Trump stated that injecting disinfectant will kill COVID in a minute?"
"Should I spray myself or my kids with disinfectant?" |
| "How frequently should facilities be cleaned to reduce the potential spread of COVID?" | "What is the survival rate of those infected by COVID who are put on a ventilator?"
"What kind of masks are recommended to protect healthcare workers from COVID exposure?"
"Will warm weather stop the outbreak of COVID?" |

Table 8: Questions from the Food and Drug Administration (FDA) and Center for Disease Control (CDC) FAQ websites that did not ask the same thing as any questions from other sources.

| Category | Example Questions |
|---|---|
| Transmission | "Can COVID spread through food?"
"Can COVID spread through water?"
"Is COVID airborne?" |
| Societal Effects | "In what way have people been affected by COVID?"
"How will COVID change the world?"
"Do you think there will be more racism during COVID?" |
| Prevention | "Should I wear a facemask?"
"How can I prevent COVID?"
"What disinfectants kill the COVID virus?" |
| Societal Response | "Have COVID checks been issued?"
"What are the steps that a hospital should take after COVID outbreak?"
"Are we blowing COVID out of proportion?" |
| Reporting | "Is COVID worse than we are being told?"
"What is the COVID fatality rate?"
"What is the most reliable COVID model right now?" |
| Origin | "Where did COVID originate?"
"Did COVID start in a lab?"
"Was COVID a bioweapon?" |
| Treatment | "What treatments are available for COVID?"
"Should COVID patients be ventilated?"
"Should I spray myself or my kids with disinfectant?" |
| Speculation | "Was COVID predicted?"
"Will COVID return next year?"
"How long will we be on lockdown for COVID?" |
| Economic Effects | "What is the impact of COVID on the global economy?"
"What industries will never be the same because of COVID?"
"Why are stock markets dipping in response to COVID?" |
| Individual Response | "How do I stay positive with COVID?"
"What are suggestions for things to do in the COVID quarantine?"
"Can I still travel?" |
| Comparison | "How are COVID and SARS-COV similar?"
"How can I tell if I have the flu or COVID?"
"How does COVID compare to other viruses?" |
| Testing | "How COVID test is done?"
"Are COVID tests accurate?"
"Should I be tested for COVID?" |
| Nomenclature | "Should COVID be capitalized?"
"What COVID stands for?"
"What is the genus of the SARS-COVID?" |
| Having COVID | "How long does it take to recover?"
"How COVID attacks the body?"
"How long is the incubation period for COVID?" |
| Symptoms | "What are the symptoms of COVID?"
"Which COVID symptoms come first?"
"Do COVID symptoms come on quickly?" |

Table 9: Sample questions from each of the 15 question categories.