# OpenReview forum: "What Are People Asking About COVID-19? A Question Classification Dataset"
_aclweb.org/ACL/2020/Workshop/NLP-COVID — NLP-COVID-2020_

### Official Review · AnonReviewer3 · 2020-06-03
**Preliminary work, limited significance**

**Rating:** 5
**Confidence:** 4

**Review:**

This paper presents a corpus of 1690 documents about COVID-19 manually annotated with questions classes, along with basic question classification algorithms.

Pros
- in scope for the conference
- dataset is publicly available
- technical details are kept to a minimum
- accessible to a large audience

Cons
- no clear statement of objectives
- no clear statement of specific contribution
- no explicit annotation guidelines (making it difficult to reproduce this effort)
- ad hoc categories (not modeled after general types of questions)
- no justification for the selection of documents
- mediocre performance of the basic question classification algorithms
- no discussion (the so-called discussion section is merely a summary)
- the claim that this dataset can help train QA systems is unwarranted at this stage

Overall
The significance of this preliminary work is extremely limited

Other comments
- the presentation of figure 2 is unnecessarily confusing

---

> ### Author Response · Authors · 2020-06-05
> **Response to review**
>
> Thanks for the review.
>
>
> - No clear statement of objectives:
>
> The objective is to release an curated dataset for developing QA/information retrieval algorithms for COVID-19.
>
> - No clear statement of specific contribution:
>
> The contribution of the paper is the carefully curated dataset.
>
> - No explicit annotation guidelines (making it difficult to reproduce this effort):
>
> Questions were annotated by matching questions that ask the same thing to a common class, and in the paper we define two questions to be asking the same thing if they can be answered with the same answer.
>
> - Ad hoc categories (not modeled after general types of questions):
>
> Categories are decided by annotators. We believe a natural breakdown of questions by topics is better than some general question types from linguistics.
>
> - No justification for the selection of documents:
>
> Publicly available documents. We selected our sources by finding trustworthy organizations that had an FAQ page about COVID-19. We exhaustively scraped questions from thirteen sources and also generated questions to have as many questions for our dataset as possible.
>
> - Mediocre performance of the basic question classification algorithms:
>
> The goal of the paper is to present the data. We benchmarked our data using a well-known off-the-shelf classifier. The goal of the paper is not to provide a state-of-the-art QA system.
>
> - No discussion (the so-called discussion section is merely a summary):
>
> Sure. The discussion section should have been named conclusions.
>
> - The claim that this dataset can help train QA systems is unwarranted at this stage:
>
> We respectfully disagree. We believe this carefully curated dataset will be of use for training/evaluating QA systems.
>
> - Overall, the significance of this preliminary work is extremely limited:
>
> We think there is value in a dataset that could potentially be used to evaluate COVID-specific NLP models. For example, in the submission immediately following ours (“An Analysis of BERT FAQ Models for COVID-19 Infobot”), the authors note that "[they] did not have COVID-19 related dataset to help [them] evaluate the performance of QA systems."

---

### Official Review · AnonReviewer1 · 2020-06-10
**A good dataset paper - some clarification and statement of limitation**

**Rating:** 7
**Confidence:** 4

**Review:**

The paper presents a dataset of questions related to COVID-19; included are annotations mapping questions to specific categories and classes.

It's always good to see papers that contribute datasets to the research community, particularly if they require significant efforts in annotation as this one does.

I think that some changes are needed in terms of positioning of this paper to make clear the contributions and benefits. The other reviewer, I believe, also struggled a little with this. I suggest an explicit "who would use this data" and "how would this dataset be used" section. I think limiting it to just people who want to train an QA IR system is a mistake. First, not all question have answers (Table 1); second, it's not obvious how the classes and categories focus, which are a large part of the paper, is core to training a QA IR system. Instead, I think the classes and categories can be used a lot more in the query intent / query understanding area of research. Pitching the paper more broadly - and being explicit on areas - would help to clarify and strengthen the contributions.

On the methodology of annotation:
- The paper rests on the technical soundness of the manual annotation process so the authors should really focus on ensuring the methodology / rigour is clear - especially as all the annotation work was done by the authors themselves.
- Categories were made up by the author. How subjective and context specific was this? What are the limitations?
- Seems like one question is mapped to one class and one category - what about overlapping questions? Surely there are questions that fall in multiple - and if there are not in this dataset then there will certainly be out in the wild. The paper, at leasts, needs to address this.
- Categories may remain static (i.e., new questions about covid-19 would likely map to existing categories). But classes will be ever evolving. For example, if a vaccine is developed then a whole host of new question classes will arise. Having these static list of classes, at this snapshot in time, risks them becoming out-of-date very quickly in a rapidly changing environment.
- Figure 2 - questions/class highly skewed. So is it actually worth having the classes? How valuable are they?
- Having an explicit limitation section at the end of the paper will really help with the above points.


Some changes in presentation could really elevate the paper:
- The definition and difference between categories and classes was not clear at the beginning of the paper. In the middle section 2, the definition finally becomes clear. A definition of both is really needed earlier in the intro.
- I found “classes” not really the best term. What about question “topics” or even question “clusters” since they are essential groups of duplicate questions. When classes are introduced, there needs to be a better justification for their purpose. It comes clearer much later but I think is needed earlier.
- Matched vs. unmatched is confusing. It’s not that they were not matched, it’s just that they were classes with only one question, right? Can a better term found? Also Table 1 presents matched/unmatched without any reference / definition - only much later in the paper do you find out.
- At some point, the paper suddenly starts talking about “labels” - it’s not clear whether this is classes or categories or both - why not just use the actual terms.

On the classifier / experiments
- The contribution of the paper is the dataset and not the ML methods. As stated, in the response to the review, the performance is merely an indication of how well current methods do on this task. The fact that the performance is good or bad does not really change the papers worth. However, insights from the experiments are part of the contribution; and these could be improved: e.g., were certain categories much harder than others? where two categories often confused for each other?

Overall, this paper provided a good dataset for the community. But it needs to be much more explicit and clear about how people might use the dataset rather than a generic reference to training QA IR system. Addressing the presentation issues and directly addressing a number of limitations will greatly improve the paper.

---

> ### Author Response · Authors · 2020-06-14
> **Review Response**
>
> It's always good to see papers that contribute datasets to the research community, particularly if they require significant efforts in annotation as this one does.
>
> - We are very grateful for the thoughtful review. See our revision in the updated PDF. (We moved some non-essential content to the supplementary materials to stay within the 4-page limit.) We see papers as the start of a conversation (rather than the end of one), and so if any new feedback arises or there are further clarifications we can make, we are happy to consider them.
>
> I think that some changes are needed in terms of positioning of this paper to make clear the contributions and benefits. The other reviewer, I believe, also struggled a little with this. I suggest an explicit "who would use this data" and "how would this dataset be used" section. I think limiting it to just people who want to train an QA IR system is a mistake. First, not all questions have answers (Table 1); second, it's not obvious how the classes and categories focus, which are a large part of the paper, is core to training a QA IR system. Instead, I think the classes and categories can be used a lot more in the query intent / query understanding area of research. Pitching the paper more broadly - and being explicit on areas - would help to clarify and strengthen the contributions.
>
> - We included a “Use cases” section in our discussion to qualify how we think our dataset might be helpful for the community. To make our mention of retrieval QA more specific, we reference an external and already deployed QA system that relates strongly to our paper. We mention query understanding and how our dataset annotations apply. And as mentioned in the last version of the paper, we emphasize that our dataset could be used for evaluation of a pre-trained COVID model, (e.g., something like Sci-BERT for COVID). Finally, we mention other related areas. If there are other use cases to add or further clarification that we can make, please let us know and we will consider them in the next revision.
>
> The paper rests on the technical soundness of the manual annotation process so the authors should really focus on ensuring the methodology / rigour is clear - especially as all the annotation work was done by the authors themselves.
>
> - Our goal was to make the collection and annotation process as clear as possible. We don’t see any specific criticism in this comment, but if we can clarify or detail the process in any way, let us know.
>
> Seems like one question is mapped to one class and one category - what about overlapping questions? Surely there are questions that fall in multiple - and if there are not in this dataset then there will certainly be out in the wild. The paper, at leasts, needs to address this.
>
> - Our first author made a new pass through the dataset and found that 40 of the 207 question clusters could arguably map to two or more categories, most frequently overlapping between the transmission and prevention categories. To make this annotation process more transparent, we came up with and posted with our dataset formal definitions of each question category so that the distinctions that we made to decide why clusters were assigned to a certain category are more transparent. We mention this in the “annotation quality” part of our paper.
>
> Categories were made up by the author. How subjective and context specific was this? What are the limitations? Categories may remain static (i.e., new questions about covid-19 would likely map to existing categories). But classes will be ever evolving. For example, if a vaccine is developed then a whole host of new question classes will arise. Having these static list of classes, at this snapshot in time, risks them becoming out-of-date very quickly in a rapidly changing environment. Figure 2 - questions/class highly skewed. So is it actually worth having the classes? How valuable are they? Having an explicit limitation section at the end of the paper will really help with the above points.
>
> - We included a limitations section that considers these three points.
>
> (continued, 1 of 2)

---

> > ### Author Response · Authors · 2020-06-14
> > **Review Response (Continued)**
> >
> > (2 of 2)
> >
> > Some changes in presentation could really elevate the paper:
> > The definition and difference between categories and classes was not clear at the beginning of the paper. In the middle section 2, the definition finally becomes clear. A definition of both is really needed earlier in the intro.
> >
> > - We defined question categories and question clusters more clearly at their first mention in the introduction.
> >
> > I found “classes” not really the best term. What about question “topics” or even question “clusters” since they are essential groups of duplicate questions. When classes are introduced, there needs to be a better justification for their purpose. It comes clearer much later but I think is needed earlier.
> >
> > - We changed our “classes” terminology to “clusters,” as per your suggestion. We made it more clear in the introduction that cluster annotations indicate questions with the same intent and could therefore be helpful in evaluating a retrieval-QA system.
> >
> > Matched vs. unmatched is confusing. It’s not that they were not matched, it’s just that they were classes with only one question, right? Can a better term found? Also Table 1 presents matched/unmatched without any reference / definition - only much later in the paper do you find out.
> >
> > - We changed this matched/unmatched terminology to “multi-question clusters” and “single-question clusters”. For clarity, we explicitly define these terms in the caption of Table 1, and again clarify again in the Single-Question Clusters paragraph that single-question clusters means questions that did not ask the same thing as any other question in the dataset. We removed all instances of “matched/unmatched questions”.
> >
> > At some point, the paper suddenly starts talking about “labels” - it’s not clear whether this is classes or categories or both - why not just use the actual terms.
> >
> > - We clarified this terminology in the revision.
> >
> > On the classifier / experiments
> > The contribution of the paper is the dataset and not the ML methods. As stated, in the response to the review, the performance is merely an indication of how well current methods do on this task. The fact that the performance is good or bad does not really change the papers worth. However, insights from the experiments are part of the contribution; and these could be improved: e.g., were certain categories much harder than others? were two categories are often confused for each other?
> >
> > - We added a confusion matrix/heatmap for the question-category classification task using the BERT+SVM. As we describe in the supplementary materials, the model did indeed confuse some categories. The biggest mistakes were between Transmission and Having COVID (cumulative 34% error rate), and between Having COVID and Symptoms (33% error rate).
> >
> > Overall, this paper provided a good dataset for the community. But it needs to be much more explicit and clear about how people might use the dataset rather than a generic reference to training QA IR system. Addressing the presentation issues and directly addressing a number of limitations will greatly improve the paper.
> >
> > - Thank you for your insightful feedback, and we hope our revised version improves on the weaknesses in the first submitted version.

---

### Official Review · AnonReviewer2 · 2020-06-29
**timely contribution, could be better positioned with regard to previous work**

**Rating:** 5
**Confidence:** 4

**Review:**

This paper presents a dataset of 1700 questions related to covid which
are hand labeled and divided into 15 general categories and 207
clusters of semantic equivalence. The value is potentially useful for
general question classification, for semantic similarity, with
particular application to reducing load on question answerers by
removing redundancy, The data set is on the small side and some
important details about how the data was collection are omitted. The
authors make a number of factual errors. All of these could be easily
corrected and the data set is a useful resource.


"we scraped questions about covid". how is a 'question about covid'
determined? Are keywords used? If so what keywords? Additionally how
were questions that had location/time-specific versions vs. questions
with only one version determined? There are number of ways this could
be done, some noisier than others, some more scalable than others.

What are all the 'synonymous ways of saying COVID?'

same answer -> same question cluster. "In which country is SARS-CoV-2
believed to have originated" and "Which country manufactures the most
face masks" have the same answer but are not the same question nor are
even related. Plenty of useful questions do not have an answer (yet);
how are these to be clustered?

"there are fewer ways to ask how long COVID will last than ways to
write a positive movie review" -- I would argue both are countably
infinite.

The cluster classification task was oddly formed. This is ultimately a
sentence similarity task or a coreference/nil clustering task. One
could use the data to pose the binary question "are these two
sentences asking the same question?" One could also posit the far more
useful "is this a question that has already been asked [and if so
which one] or is this a novel question?" task of coref/nil
clustering. Static assignment to clusters seems wrong for that kind of
data. By specifically excluding clusters with a small number of
questions you specifically skirt the issue you would have to deal with
in real application of this data.

---

> ### Author Response · Authors · 2020-06-30
> **Thank you for the review**
>
> > This paper presents a dataset of 1700 questions related to covid which are hand labeled and divided into 15 general categories and 207 clusters of semantic equivalence. The value is potentially useful for general question classification, for semantic similarity, with particular application to reducing load on question answerers by removing redundancy, The data set is on the small side and some important details about how the data was collection are omitted. The authors make a number of factual errors. All of these could be easily corrected and the data set is a useful resource.
>
> Thank you for the insightful review. We plan to make an update to the manuscript soon (in the next couple of days), and hope that it will be a stronger paper after these revisions.
>
> Our dataset is indeed small, and we did everything we could given the circumstances at the time (our paper was submitted more than one month ago)---we scraped from all the official FAQ sources we could find and collected hundreds of questions from search engines. The call for papers for this workshop specifically asks for “active and late-breaking research”. With more time, we can obviously grow the dataset, what we present here is part of an ongoing project to collect a highly quality dataset of questions related to COVID-19. We have made the dataset available since the beginning to help with research in this area.
>
> > "we scraped questions about covid". how is a 'question about covid' determined? Are keywords used? If so what keywords? Additionally how were questions that had location/time-specific versions vs. questions with only one version determined? There are a number of ways this could be done, some noisier than others, some more scalable than others.
>
> For sources such as the CDC and FDA, we used all the questions on their COVID FAQ website. For sources such as Google, Bing, Yahoo, and Quora, we used the keywords  “COVID” and “coronavirus” to find COVID-related questions. We will specify this point in the revision of our paper.
>
> All location/time-specific questions were manually determined by the first and last author as such. For even more clarity, involved readers can refer to the original sources that we scraped---these were uploaded to our Github at the time of submission.
>
> > What are all the 'synonymous ways of saying COVID?'
>
> In this dataset, we defined [“SARS-COV-2”, “2019-nCOV”, “coronavirus”, “COVID-19”, “COVID19”] as synonymous ways of saying COVID. Based on manual examination, these covered most of the dataset.
>
> > same answer -> same question cluster. "In which country is SARS-CoV-2 believed to have originated" and "Which country manufactures the most face masks" have the same answer but are not the same question nor are even related. Plenty of useful questions do not have an answer (yet); how are these to be clustered?
>
> Thank you for making this clarifying comment. In practice, most of the questions were not one-word answers---i.e., answers were typically like “Early on, many of the patients at the epicenter of the COVID-19 outbreak in China had some link to a large seafood and live animal market, suggesting animal-to-person spread”, not simply “China.”
> Rather, what we meant was that, for example, the answer “COVID-19 originated in Wuhan, China.” responds to both “Where is SARS-CoV-2 believed to have originated” and “Where did COVID-19 start”, and we would therefore put these two questions in the same cluster.
>
> Whether two questions belonged to the same cluster was decided in most cases by the annotator, without referring to the answers of the questions. We realize that there is some subjectivity here, as is the case with ground-truth labels of many datasets. We think many annotators would agree with most of our dataset’s annotations, as we showed with our quality control checks with independent annotators.
>
> > "there are fewer ways to ask how long COVID will last than ways to write a positive movie review" -- I would argue both are countably infinite.
>
> In this sentence, we hoped to provide some intuition about why data augmentation helped, without intending to give a formal proof. We see your point; we have rephrased it in the revised version.

---

> > ### Author Response · Authors · 2020-06-30
> > **Question clustering/similarity model**
> >
> > > The cluster classification task was oddly formed. This is ultimately a sentence similarity task or a coreference/nil clustering task. One could use the data to pose the binary question "are these two sentences asking the same question?" One could also posit the far more useful "is this a question that has already been asked [and if so which one] or is this a novel question?" task of coref/nil clustering. Static assignment to clusters seems wrong for that kind of data. By specifically excluding clusters with a small number of questions you specifically skirt the issue you would have to deal with in real application of this data.
> >
> > Thanks for this insightful comment. We aimed to provide a straightforward evaluation for matching questions that asked the same thing, but your perspective and framing of this problem is probably more comprehensive.
> >
> > In the coming days, we will reframe this task to use all data (including single-question clusters), and ask our model to identify whether a given question has already been asked in the database. Specifically, we will split the data by class into 70% train and 30% test, and then for each testing question, our model will predict whether that question was similar enough to an existing question in the database (and if so which one) or if it was a novel question. We will then report performance using a single number metric that captures whether this prediction was made correctly.
> >
> > As this is a substantive change that requires re-running our models and evaluation, we are already working and hope to get this revision back within the next few days.

---

### Decision · Program_Chairs · 2020-10-15

**Decision:**

Accept

**Comment:**

Formally recording the decision on the revision previously communicated.